# How Warburg-Associated Lactic Acidosis Rewires Cancer Cell Energy Metabolism to Resist Glucose Deprivation

**DOI:** 10.3390/cancers15051417

**Published:** 2023-02-23

**Authors:** Zoé Daverio, Aneta Balcerczyk, Gilles J. P. Rautureau, Baptiste Panthu

**Affiliations:** 1Laboratoire CarMeN, Institut National de la Santé et de la Recherche Médicale U1060, Université Claude Bernard Lyon 1, 69100 Lyon, France; 2Department of Oncobiology and Epigenetics, Faculty of Biology and Environmental Protection, University of Lodz, 90-236 Lodz, Poland; 3Institut de Chimie et Biochimie Moléculaires et Supramoléculaires, Université Claude Bernard Lyon 1, 69622 Lyon, France

**Keywords:** lactic acidosis, glucose deprivation, tumour heterogeneity, metabolic symbiosis, Warburg effect

## Abstract

**Simple Summary:**

Lactic acidosis is a prominent feature of the tumour microenvironment and a key player in cancer metabolism. This review is aimed at combining the mechanisms through which lactic acidosis alters the metabolism of cancer cells, and determining how this effect could bring valuable contribution to the current understanding of the metabolism of whole tumours. This work also highlights the therapeutic perspectives that advances in lactic acidosis understanding open up.

**Abstract:**

Lactic acidosis, a hallmark of solid tumour microenvironment, originates from lactate hyperproduction and its co-secretion with protons by cancer cells displaying the Warburg effect. Long considered a side effect of cancer metabolism, lactic acidosis is now known to play a major role in tumour physiology, aggressiveness and treatment efficiency. Growing evidence shows that it promotes cancer cell resistance to glucose deprivation, a common feature of tumours. Here we review the current understanding of how extracellular lactate and acidosis, acting as a combination of enzymatic inhibitors, signal, and nutrient, switch cancer cell metabolism from the Warburg effect to an oxidative metabolic phenotype, which allows cancer cells to withstand glucose deprivation, and makes lactic acidosis a promising anticancer target. We also discuss how the evidence about lactic acidosis’ effect could be integrated in the understanding of the whole-tumour metabolism and what perspectives it opens up for future research.

## 1. Introduction

Lactic acidosis is a hallmark of the tumour microenvironment, one that has been shown to promote cancer resistance to chemotherapy [1]. It results from the intensive secretion of lactate and protons in the presence of glucose by cells displaying the Warburg effect, a characteristic anomaly of proliferating, and, particularly, cancer cells. Cells harbouring the Warburg effect perform high-rate glycolysis, lactic fermentation, and co-excretion of lactate and protons [2]. This enables them to proliferate at a high rate in the presence of glucose, which they consume avidly. However, the rapid consumption of glucose leads to its exhaustion, and an energetic dead-end and paradox. Interestingly, lactic acidosis has been shown to help cancer cells withstand glucose deprivation [3]. In media conditioned with high lactate concentration and acidity, cancer cell lines avoid apoptosis and survive 10 times longer in the absence of glucose. Further studies have demonstrated that cancer cells resist glucose starvation by reprogramming their metabolism [4,5]. In this review, we focus on the essential literature addressing how lactic acidosis affects energy metabolism and preserves homeostasis in glucose-deprived cancer cells, and what therapeutic prospects it opens up. We then discuss how this effect at the cellular scale could help understand the metabolism of whole tumours.

## 2. Defining the Experimental Conditions of the Presented Studies

In this work, we review a series of studies relevant to address how lactic acidosis helps cells resist glucose deficiency. These studies are performed in varying conditions (Table 1). In order to clarify the various experimental conditions, we emphasise the following definitions [3]. “Lactosis” refers to an in vitro condition in which extracellular lactate concentration exceeds 15 mM. It must be noted that most presented studies were performed with culture media containing 10% foetal bovine serum, which brings ~1.5 mM lactate to the medium [6]. At pH 6.7, >15 mM added lactate helps cancer cells resist glucose deprivation [3]. “Acidosis” refers to an extracellular pH of 5.8–6.7. Under pH 6.7 normal cells suffer from acidosis, and tumour pH can drop down to 5.8 [4]. ”Lactic acidosis” refers to the combination of both lactosis and acidosis. Lactic acidosis and acidosis are frequently encountered in tumours [7]. Both originate from the co-secretion of lactate and protons, and acidosis is also caused by the mitochondrial production of CO_2_ and its dissociation into HCO_3_^−^ and H^+^ [8]. Lactosis is a condition virtually absent in vivo, but one that can be achieved easily in vitro to study the effect of lactate independently from acidification by adding buffered sodium lactate to the medium.

“Glucose deprivation” or “depletion” refers to conditions where glucose is scarce, but not necessarily absent from the milieu. Intratumoral glucose concentration can drop to 0.1–0.4 mM, while its level in healthy tissues is ~1 mM [9]. In vitro studies recreate glucose deprivation with culture media that contain, initially, up to 3 mM glucose, the amount that cancer cells typically deplete in one day [3,6].

**Table 1 cancers-15-01417-t001:** The presented studies addressing lactic acidosis’ impact on cell energy metabolism are performed under various conditions. For each reference, the tested cell line or cancer type and medium conditions (glucose concentration, lactate concentration, and pH) are specified. When unspecified, the pH value was assumed to equal 7.4.

Reference	Glucose Concentration (mM)	Lactate Concentration (mM)	pH	Cell Lines or Tumour Origin
[3]	3	20	6.7	4T1, Bcap37, RKO, SGC7901
[10]	Unspecified	25	6 to 6.7	HMEC, DU145, SiHa, WiDr
[4]	10	10	6.5	MCF-7, MDA-MB-468, MDA-MB-231, SkBr3
[11]	5 and 25	5 to 30	6.7	U251 and glioblastoma
[12]	Unspecified	10 or 20	7.4	A549, H1299
[13]	10	5 to 30	7.4	A549, H1299
[14]	Unspecified	10 or 30	7.4	SiHa and mouse xenograft
[5]	6	25	6.5	4T1, Bcap37, HeLa, A549
[15]	Unspecified	4 to 40	5 to 8	MCF7, T47D
[16]	Unspecified	5 or 10	7.4	A549, H1299, BEAS-2B
[17]	10	3 to 40	6.2	A549, A427, MCF7, MRC5
[18]	Unspecified	0	6.5	A549, H1299, MRC5
[19]	5	10	7.4	SiHa, HeLa
[20]	10	10 or 25	6.7	MCF-7, ZR-75-1, T47D, MDA-MB-231, MDA-MB-157
[21]	5.6	10 or 20	6.7	LS174T, HCT116, MCT4
[22]	Unspecified	20	7.4	MCF7
[23]	Unspecified	10	7.4	MDA-MB-231
[24]	0	28	6.2	A549, A427
[25]	Unspecified	20	7.4	U87-MG, A172, U251
[1]	Unspecified	20	7.4	92.1
[26]	0	10	7.4	MDA436 and mouse xenograft
[27]	10	2 to 20	7.4	Human myeloid cell lines
[28]	0.175	4	7.4	glioma stem cells
[29]	2.5 or 25	10	7.4	Colo205, Ls174T, Mosers, HT29
[30]	1 to 2.5	25	7.4	MCF-7
[31]	Unspecified	20	7.4	Huh-7, Hep3B
[32]	0	20	6.8	A549
[33]	0	20	6.7	4T1, HeLa, NCI–H460
[34]	Unspecified	0	6.5	PANC-1, SW1990
[35]	Unspecified	12	6.8	PaTu-8902, HeLa, HepG2, HDF

## 3. Lactic Acidosis Seen by Cancer Research: A Brief History

In the 2000s, cancer research took a renewed interest in the Warburg effect, a hallmark of cancer discovered a century ago [2,36,37]. As a consequence, views on lactic acidosis changed drastically.

Acidosis had been known to promote tumour aggressiveness by exerting a selective pressure. Some cancer cells had been shown to survive acidosis by maintaining an alkaline intracellular pH, while other cells—cancerous or healthy—underwent hydrolysis and death [38,39,40]. The proliferation of those selected cells, which are more resistant to unfavourable environments, had been known to increase tumour malignancy [41,42]. As for lactate, it had been considered more of a by-product of glycolysis until the 1980s, when its use as a nutrient in non-cancerous tissues was discovered [43,44]. The role of extracellular lactate in cancer was investigated only later, in the 2000s [45,46], when it was found to correlate with tumour malignancy [47,48,49,50]. Two explanations for this were initially proposed. First, lactate promotes relaxation of the tissue surrounding the tumour, which would make room for its development and metastasis [48]. Second, lactate makes the cellular environment hostile, as does acidosis [38], which promotes angiogenesis [47].

The metabolic importance of extracellular lactate and lactic acidosis was first evidenced in 2008. Lactic acidosis was shown to alter the expression of metabolism genes [10] and, more importantly, lactate was proven to be, per se, a key source of energy for cancer cells [51]. In 2009, the term “reverse Warburg effect” was first used to describe cancer cells not showing the Warburg effect, but instead inducing it in neighbouring stromal fibroblasts and consuming the lactate produced by them [52]. These discoveries reappraised the paradigm of the Warburg effect, showing that it wasn’t compulsory in cancer since lactate could be metabolised rather than only produced. Following these works, in 2012, Wu et al., demonstrated that lactic acidosis allows cells to avoid glucose starvation [3]. Lactic acidosis rescues glucose-deprived cancer cells, but importantly, acidosis or lactosis alone have much more limited effects. After this pioneering work, lactic acidosis was further shown to reprogram cell metabolism [5]. Nowadays, extracellular lactate and acidosis are viewed as central players in cancer cell metabolism [53,54,55].

## 4. Lactic Acidosis’ Effect on Energy Metabolism

Lactic acidosis was shown to impact numerous aspects of energy metabolism. We focus here on nutrient import, glycolysis, the tricarboxylic acid (TCA) cycle, oxidative phosphorylation (OxPhos), and pathways generating reduced coenzymes (Figure 1).

### 4.1. Lactic Acidosis and Exchanges at the Plasma Membrane

In glucose deprivation, the capacity of cancer cells to uptake and metabolise alternative nutrients is key to their survival [56]. Extracellular acidosis and lactosis were shown to increase such capacity.

#### 4.1.1. Acidosis Sustains the Activity of Proton-Nutrient Symporters

Extracellular acidosis has a direct impact on exchanges at the plasma membrane [57]. In healthy tissues, protons are more concentrated inside the cell than outside. In tumours, the contrary is true [58,59]. Extracellular acidosis inverts the transmembrane proton gradient in tumour cells, which may positively impact proton-nutrient symports. Of interest, lactate is imported in cancer cells via the monocarboxylate transporters (MCTs) [51]. Since MCTs co-transport lactate with a proton, lactate import should be sensitive to the proton gradient and facilitated under acidosis. This mechanism is expected to explain why cancer cells respond differently to lactosis and lactic acidosis [3], since a rise in extracellular lactate only increases intracellular lactate levels in acidic conditions [60]. The co-transport of extracellular lactate and protons probably underlies the synergy of their effects on intracellular metabolism (Figure 1).

Of note, protons are also co-imported with several other nutrients, such as Fe^2+,^ folates, amino acids and peptides. Brown & Ganapathy suggested that acidosis may affect their uptake (Figure 1), but this hypothesis remains to be confirmed [61].

#### 4.1.2. Lactate and Acidosis Indirectly Enhance Nutrient Uptake

Extracellular lactic acidosis indirectly promotes the uptake of several nutrients (Figure 1). The import of lactate itself is increased in lactic acidosis, in part due to MCT1 overexpression [11], which is a response to an extracellular lactate signal [62] that is potentiated by extracellular acidity [12]. Extracellular lactate induces MCT1 and MCT4 via the G-protein-coupled receptor 81 (GPR81) transduction pathway [62]. In acidosis, extracellular lactate can also induce GPR81 expression [12,13]. Extracellular lactate signal enhancing MCT-mediated lactate import is necessary to cancer cell survival in absence of glucose, glutamine and pyruvate [62]. This supports the idea that ‘lactate induces its own metabolism’, which doesn’t exclude other regulations of MCT expression [63].

Glutamine uptake is increased by extracellular lactate or acidosis, as both conditions increase the expression of the glutamine transporter ASCT2 (alanine, serine and cysteine transporter 2) [14,64]. Fatty acid uptake is enhanced by acidosis [65], and folate import is intensified by 10 mM extracellular lactic acid [15].

Finally, and importantly, lactic acidosis seems to minimise glucose uptake, but not in all cell lines and cancers. Lactic acidosis decreases glucose uptake in various cell lines [5], as lactate in lung cancer cell lines [16]. On the opposite, 10 mM lactate has no effect on glucose uptake in the T47D breast cancer cell line [15]. The expression of the glucose transporters GLUT1 and GLUT4 are decreased by 2 mM lactate and acidosis in lung and breast cancer cell lines [17], and by acidosis in cervix, pharynx and colon cancer cell lines [64] but not in lung cancer cell lines [18].

#### 4.1.3. Lactic Acidosis and pH Homeostasis

Cell exposure to lactic acidosis is associated with a drop in intracellular pH from 7.3 to ~6.9 [5]. Behind this acidification, several probable effects may be discerned. On the one hand, as discussed earlier, acidosis enhances proton-nutrient co-import, which could contribute to cellular acidification. On the other hand, lactate as a signal can mitigate the drop in pH by favouring alkalinization. A level of 10 mM extracellular lactate induces Carbonic Anhydrase IX (CA IX) [19], a transmembrane enzyme supporting proton export and a key regulator of cell pH [66] (Figure 1).

### 4.2. Lactic Acidosis, Glycolysis, and Lactic Fermentation

Unlike cells showing the Warburg effect, in which glycolysis and lactate dehydrogenase (LDH)-catalysed lactic fermentation are known to be hyperactive, cells exposed to lactic acidosis show a reduction in these pathways’ activity.

In glucose abundance, acidosis and lactic acidosis lower glucose consumption and lactate secretion [4,5,20], which indicates that glycolysis and lactic fermentation are downregulated. More interestingly, lactic acidosis decreases cancer cell dependency on glucose catabolism [1]. Thus, in glucose sufficiency, lactic acidosis minimises glucose catabolism activity and its importance in cell survival (Figure 1).

Glycolysis and lactic fermentation are likely downregulated at the level of both gene expression and enzyme activity. The expression of glycolysis enzymes is reduced by lactic acidosis in breast cancer cell lines [10], and by extracellular lactate in lung cancer cell lines [16], but it is maintained by extracellular lactate in breast cancer cell lines [21,22]. The activity of glycolysis enzymes, especially the rate-limiting hexokinase and phosphofructokinase [67], is directly decreased by intracellular acidification [4] (Figure 1). In line, intracellular acidification has been predicted in silico to hinder the Warburg effect [68]. Intracellular lactate accumulation, in parallel, directly inhibits lactic fermentation [5] (Figure 1). The interconversion of lactate and pyruvate through LDH follows the mass action law, therefore a rise in lactate concentration inhibits its production from pyruvate and favours the reverse reaction. This thermodynamic effect leads to a complete stop of lactic fermentation at ~25 mM intracellular lactate [5]. This concentration is within the range resulting from lactic acidosis.

### 4.3. Lactic Acidosis and Mitochondrial Catabolism

#### 4.3.1. Lactic Acidosis Intensifies Mitochondrial Catabolism

Lactic acidosis enhances mitochondrial metabolic activity, in particular the TCA cycle and OxPhos. Both lactic acidosis and lactosis enhance mitochondrial biogenesis [23,24] and the expression of the enzymes of the TCA cycle and OxPhos [11,25], which potentiates mitochondrial catabolism and ATP production. The reactivation of those pathways allows the maintenance of the cellular ATP concentration in glucose deprivation and increases resistance to starvation [11].

#### 4.3.2. Lactic Acidosis Shapes TCA Cycle Alternative Fueling

In addition to glucose-derived pyruvate, the TCA cycle can be supplied with various substrates. This flexibility is particularly true of cancer cells [69]. In challenging nutritional contexts such as glucose deprivation, the TCA cycle of cancer cells can be sustained by alternative nutrients. Lactate and glutamine are its main substrate suppliers after glucose [70]. The use of both is promoted by lactic acidosis.

The pyruvate generated from lactate can directly sustain the TCA cycle [26,27,28,29,30,31,71] (Figure 1). This pathway depends on upstream lactate import by MCTs, whose enhancement in lactic acidosis is discussed in Section 4.1.2. In line, extracellular lactate increases the mitochondrial membrane potential, and hence ATP production efficiency in OxPhos [21,72], and could even be necessary to pro-tumoural cell proliferation [32]. In more detail, the routing of lactate to mitochondria is debated. In the classical view, lactate is converted to pyruvate in the cytosol, then pyruvate is shuttled to mitochondria [73,74]. In addition to this classical way, Brooks et al. proposed an alternative model in which lactate would be shuttled to mitochondria via the mitochondrial lactate oxidation complex (mLOC), that includes MCT1 [75]. The controversy raised by this model has been well-reviewed in [76,77], that summarized the evidence for and against it in non-cancer cells. In cancer cells supplied with sufficient glucose, lactate’s contribution to the TCA cycle over glucose remains under debate: some studies suggest that lactate shuttled to mitochondria is preferred [71], while others question this [78]. Either way, under glucose deprivation, we can hypothesise that lactate’s contribution to the TCA cycle is of significant importance.

Glutamine is a major nutrient for cancer cells. It undergoes oxidative glutaminolysis in mitochondria, where it is processed by glutaminase 1 or 2 (GLS1/2) and then glutamate dehydrogenase 1 (GDH1) to sustain the TCA cycle. Lactic acidosis [20], acidosis [20,64], and lactate [14] upregulate GLS1 and GLS2 and stimulate oxidative glutaminolysis. Lactic acidosis, however, doesn’t necessarily promote glutamine consumption compared to lactosis [23]. In summary, either extracellular lactate, acidosis or lactic acidosis enhance glutamine utilisation by inducing glutaminase expression (Figure 1).

### 4.4. Lactic Acidosis and Redox Homeostasis

Cell survival requires redox homeostasis, i.e., controlled levels of reactive oxygen species (ROS) and redox coenzymes. The former lead to cell death when they accumulate, and the latter support the entire metabolism and cellular antioxidant defences.

Particularly, a high NADPH/NADP^+^ ratio kinetically favours anabolic reactions and helps keep ROS levels low. In cancer cells this ratio is abnormally high and sustains hyperactive anabolism [70]. High NADPH levels are supported by the oxidation of nutrients, such as lactate and glutamine via the TCA cycle and then oxidation of glutamine- and lactate-derived malate and isocitrate by the malic enzyme 1 (ME1) and Isocitrate Dehydrogenase 1 (IDH1), and mainly glucose via the pentose phosphate pathway (PPP). Redox homeostasis in cancer cells is therefore particularly sensitive to nutritional stress such as glucose deprivation.

In this condition, lactic acidosis helps stabilise the NADPH/NADP^+^ ratio at ~50% of its level in glucose sufficiency [3]. Likely, the gatekeepers of the NADPH/NADP^+^ ratio in glucose abundance have reduced efficiency under glucose deprivation and lactic acidosis, whereas new control mechanisms gain importance. On the one hand, glutamine use via ME1 is not necessary to the maintenance of the NADPH/NADP^+^ ratio under lactic acidosis [33]. Glutamine would indeed be completely degraded in mitochondrial catabolism instead of sustaining ME1 activity [20]. On the other hand, the glucose directed away from glycolysis towards the PPP would prevail more in NADPH/NADP^+^ maintenance under lactic acidosis. Lactic acidosis [20] and acidosis [34] respectively increase the expression and activity of glucose-6-phosphate dehydrogenase (G6PD), the first enzyme of the PPP, and lactic acidosis makes G6PD activity necessary to NADPH/NADP^+^ ratio maintenance and cell survival [20] in glucose sufficiency. However in glucose deprivation, the PPP alone cannot maintain redox balance [33]. Alternatively, lactate would become a key player in NADPH/NADP^+^ ratio maintenance, via the TCA cycle [35], and IDH1 [33].

Whether, in glucose abundance, such reprogramming strengthens cell defences against ROS level increase is uncertain. Acidosis increases ROS levels [35] and cell sensitivity to oxidative stress, but cell adaptation to acidosis decreases them [34]. Lactate import through MCT1 is key to maintain low ROS levels [79]. Lactic acidosis was found to either increase ROS levels, as does acidosis [20], or to rescue acidosis’ negative effect [35]. At any rate, in glucose deprivation, lactic acidosis mainly prevents increased ROS levels by providing IDH1 with its substrate [33].

A high NADH/NAD^+^ ratio supports ATP production. Lactic acidosis impact on the NADH/NAD^+^ ratio has not been directly investigated. However lactate use by the TCA cycle increases the NADH/NAD^+^ ratio in glucose deprivation [30]. This increase could contribute to the inhibition of glycolysis by lactic acidosis: a high NADH/NAD^+^ ratio would inhibit glycolysis according to the mass action law. Yet this hypothesis remains to be tested.

### 4.5. Section Summary

In the energy metabolism of cancer cells, acidosis and extracellular lactate act as enzymatic inhibitors, and lactate as a signal and a nutrient. They mostly curb glycolysis and lactic fermentation and enhance the TCA cycle and OxPhos (Figure 1). Acidification and lactate accumulation in the tumour microenvironment would promote and sustain an oxidative phenotype, which is fitter than the fermenting phenotype in glucose deprivation, an adverse nutritional context that is common in tumours.

## 5. Therapeutic Strategies Targeting Lactic Acidosis

Nowadays, lactic acidosis *per se* is targeted in therapies directed against cancer. Of note, it is also a major target in the treatment of type 2 diabetes [80,81]. Neutralising acidosis in tumours has been proposed as a way to restore sensitivity of cancer cells to glucose starvation and increase the efficacy of regular treatments. The proof of principle of this approach has been established by combining transarterial chemoembolization (TACE) with the infusion of bicarbonate, a basifying agent that turns neoplastic lactic acidosis into lactosis [82,83]. Compared to TACE alone, TILA-TACE (Targeting-Intratumoural-Lactic-Acidosis TACE) presented a very significantly enhanced anticancer activity for patients with hepatocellular carcinoma. The mechanisms underlying this activity have been evaluated in detail by Ying et al. [84].

Modulating extracellular lactate availability in tumours by nanomedicine is another promising therapeutic strategy. The delivery by nanoparticles of a cocktail of lactate oxidases and catalases to colon carcinoma cells in vitro suppresses tumoural lactosis and stops cell proliferation [85]. The delivery by nanoparticles of a glucose catalase combined with a MCT1 inhibitor, that together prevent the use of both glucose and lactate by tumour cells, inhibits the proliferation of SiHa cell line xenografts in mice [86]. Conversely, lactate-loaded nanoparticles induce an overload of lactate and cytotoxicity in orthotopic glioblastoma models, although only in normoxic conditions and not in hypoxia [87]. 

Understanding the effects of lactic acidosis also helps reappraise the potential of already-existing targets. In particular, the lactate transporter MCT1 was formerly targeted to inhibit lactate secretion in cells showing the Warburg effect, and is now targeted to hamper lactate uptake [27,88,89,90,91]. Similarly the strategies targeting LDH isoforms were aimed historically at inhibiting lactate production from pyruvate. The LDHA isoform, that has a higher affinity for pyruvate than for lactate and catalyses preferentially lactate production, is a historical target that still attracts much attention [92]. However, with the discovery of lactic acidosis effect, the LDHB isoform that catalyses preferentially the conversion of lactate to pyruvate now rises as an alternative target [32].

## 6. Implications of Lactic Acidosis in the Whole-Tumour Metabolism

Deciphering how lactic acidosis impacts cancer cells enlightens important aspects of the metabolism of the whole tumour, and raises new perspectives to complement its understanding.

From the belief that cancer cells have a unique metabolic signature, i.e., the Warburg effect, research has progressively recognized intratumoural heterogeneity as the metabolic hallmark of cancer [55]. The main metabolic heterogeneity in tumours is now suggested to be mitochondrial activity [93,94], that is promoted and sustained by lactic acidosis. To describe this heterogeneity, tumours have traditionally been modelled as the coexistence of two metabolic populations: oxidative cells relying on OxPhos and fermenting cells showing the Warburg effect and relying on glycolysis and lactic fermentation [17,28,51,91,95,96]. Oxidative cells would be located in normoxic regions, in perivascular compartments [7,11,97], and fermenting cells in hypoxic regions farther from blood vessels [11,51,97] (Figure 2). Each population would thrive on different energy sources, fermenting cells glucose and oxidative ones lactate, and lactate would be transferred from fermenting to oxidative cells [95]. This model is supported by the coexistence in tumours of cells overexpressing MCT4, a preferential lactate exporter, and cells overexpressing MCT1, a preferential lactate importer [11,51,95,98]. Interestingly, a possible lactate transport via gap junctions has been evidenced recently [99,100,101]. This lactate transfer supports the idea of a metabolic symbiosis between both populations within tumours [17,28,46,51,61,91,95,96,101,102,103]. In this model, a central question is how the metabolic phenotypes of both populations are determined [46]. Hypoxia is thought to be the major promoter of the fermenting phenotype [102,104]. Lactic acidosis, according to the evidence presented in this work, is likely the promoter of the oxidative phenotype [5,11].

However this hypothetical scenario raises a paradox: the oxidative phenotype, that derives from lactic acidosis, i.e., from the fermenting phenotype that is promoted by hypoxia, cannot thrive in hypoxic conditions. Two hypotheses could solve this paradox. In the first hypothesis, fermenting cells would induce the oxidative phenotype in their neighbours, located in better-perfused regions. However lactic acidosis intensity, maximal around secretory cells, decreases with the distance [51,105], which raises the question of the minimal level of lactic acidosis necessary to promote the oxidative phenotype. In the second hypothesis, lactic acidosis would feedback the Warburg effect in fermenting cells by switching them to an oxidative phenotype, which questions the minimal oxygen level necessary for the oxidative phenotype to survive. A possible answer to this question is that in the meantime, lactic acidosis could promote angiogenesis [38,47,61]. This questions the timeline of lactic acidosis action, in the promotion of both oxidative phenotype and angiogenesis. A third perspective to answer the paradox is to address how, earlier, hypoxia and lactic acidosis may interplay in the promotion of metabolic phenotypes, which has caught little attention until now [18,20,23,106].

## 7. Conclusions

Lactic acidosis associated with tumour progression allows cancer cells to survive in unfavourable environments. In the last decade, the influence of neoplastic lactic acidosis on the energy metabolism of cancer cells has been deciphered. Lactic reduces glycolysis and lactic fermentation, stimulates the TCA cycle and OxPhos, and promotes the use of alternative nutrients. All in all, it contributes to cell resistance to glucose deprivation. Cancelling lactic acidosis’ effect is therefore a relevant anticancer strategy that restores cancer cell sensitivity to glucose deprivation, a common feature of the tumour microenvironment. In the future, clarifying how lactic acidosis action is integrated in the whole-tumour metabolism would be of high interest.

## Figures and Tables

**Figure 1 cancers-15-01417-f001:**
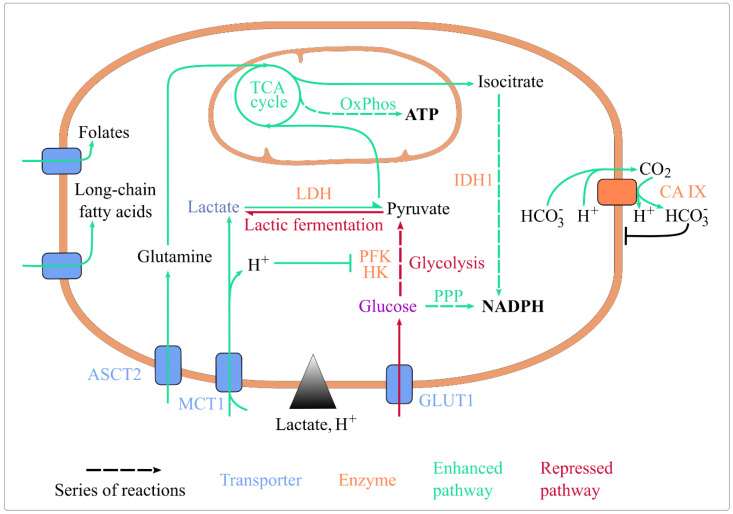
Lactic acidosis rewires energy metabolism and maintains cellular homeostasis. Lactic acidosis enhances the uptake of folate, long-chain fatty acids, glutamine, and lactate. It represses glucose import, glycolysis (by inhibiting HK and PFK, its rate-limiting enzymes) and lactic fermentation. It enhances lactate conversion to pyruvate, routing of pyruvate and glutamine towards the TCA cycle, ATP generation by OxPhos, and coenzyme reduction by IDH1 and the oxidative PPP. It also upregulates CA IX expression, which basifies intracellular pH. Abbreviations: ASCT2: Alanine, Serine, Cysteine Transporter 2; CA IX: carbonate anhydrase 1; GLUT1: glucose transporter 1; HK: hexokinase; IDH1: isocitrate dehydrogenase 1; MCT1: monocarboxylate transporter 1; OxPhos: oxidative phosphorylation; PFK1: phosphofructokinase 1; PPP: pentose phosphate pathway; TCA: tricarboxylic acid.

**Figure 2 cancers-15-01417-f002:**
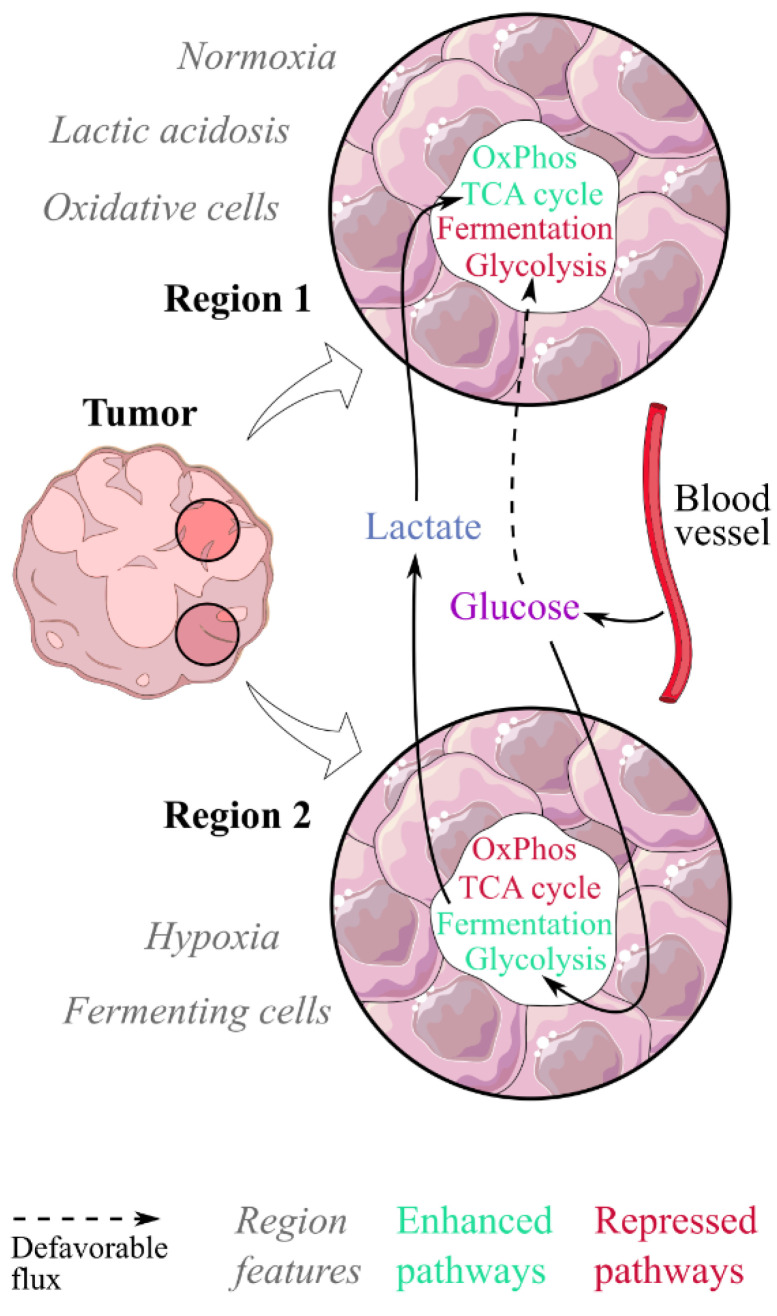
Lactic acidosis would contribute to a metabolic symbiosis between fermenting and oxidative cells within tumours. In this model, two populations coexist in tumours: fermenting cells in hypoxic regions, and oxidative cells in normoxic regions where lactic acidosis would exert its effect. Fermenting cells would consume the glucose spared by oxidative cells and generate the lactate fueling them, both being in a metabolic symbiosis. Lactic acidosis promotes the switch from a fermenting to an oxidative phenotype.

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
