# Peer review of "How Warburg-Associated Lactic Acidosis Rewires Cancer Cell Energy Metabolism to Resist Glucose Deprivation"

_cancers, 2023, doi:10.3390/cancers15051417_

Round 1
Reviewer 1 Report
In current manuscript ‘How Warburg-associated lactic acidosis rewires cancer cell energy metabolism to resist glucose deprivation’, the authors have tried to demonstrate the impact of lactic acidosis on the metabolic profile of cancer cells and how does increased lactate shape the metabolism of cancer cells.
The manuscript is well written and have demonstrated the idea in a comprehensive manner with beautiful illustrations incorporated in the manuscript. However, authors are advised to look at some of the points mentioned below.
In section 4.3.2 authors have demonstrated lactate role in TCA. Discussing the impact of MCT1/4 in metabolic shift of cancer cells in this section in a more detailed fashion is recommended.
Line 111 is missing reference, please add relevant reference
Lactic acidosis and acidosis are terms generally used synonymously. The authors have defined both the terms. Is there any other source in cellular metabolism involved in acidosis other than lactic acid? Please explain?
Author Response
The referee has raised three relevant points in their suggestions, and we hope that the following modifications improve the quality of the work.
To emphasize the role of MCT1/4 transporters in lactate metabolism in mitochondria, we added the following elements:
1)-line 152: "This supports the idea that 'lactate induces its own metabolism', which doesn't exclude other regulations of MCT expression (Payen 2020)."
-line 211: "This pathway depends on upstream lactate import by MCTs, whose enhancement in lactic acidosis is discussed in section 4.1.2."
and to precise the possible role of MCTs in mitochondrial lactate import, we replaced
-line 214 : "In glucose abundance, lactate's contribution to the TCA cycle over glucose remains under debate: some studies suggest that lactate shuttled to mitochondria is preferred [58], while others question this [66]."
by: "In further detail, the routing of lactate to mitochondria is debated. In the classical view, lactate is converted to pyruvate in the cytosol, then pyruvate is shuttled to mitochondria (Herzig 2012, Bricker 2012). In addition to this classical way, Brooks et al. proposed an alternative model in which lactate would be shuttled to mitochondria via the mitochondrial lactate oxidation complex (mLOC), that includes MCT1 (Hashimoto 2006). The controversy raised by this model has been well-reviewed in (Li 2022, Brooks 2018), that summarized the evidence for and against it in non-cancer cells. In cancer cells supplied with sufficient glucose, lactate's contribution to the TCA cycle over glucose remains under debate: some studies suggest that lactate shuttled to mitochondria is preferred [58], while others question this [66]."
2) Line 119, we added the following reference: Elingaard-Larsen 2020: doi: 10.1007/112_2020_23
3) In order to clarify the origins of acidosis and lactic acidosis in tumours, we replaced :
line 63: 'Acidosis and lactic acidosis are frequently encountered in tumours [6], where they result from the secretion of either protons or lactic acid.'
by: ''Lactic acidosis and acidosis are frequently encountered in tumours [6]. Both originate from the co-secretion of lactate and protons, and the lattest is also caused in oxygenated tumour regions by the mitochondrial production of CO2 and its dissociation into HCO3- and H+. Both processes occur at a high rate in cancer cells and sustain their proliferation (Corbet & Feron 2017).'
Reviewer 2 Report
The effect of lactic acidosis in the microenvironment of solid tumors is discussed in the review article. The Warburg effect in cancerous cells leads to an excessive lactic acid synthesis and secretion, which causes lactic acidosis. It is now understood to have a substantial role in the physiology of tumors, their aggressiveness, and the success of treatments. Recent research has shown that lactic acidosis encourages cancer cell resistance to glucose deprivation and aids in the transition from the Warburg effect to an oxidative metabolic phenotype. In this article, the future possibilities of study in this area are discussed, along with how lactic acidosis may be explored as a target for cancer therapy.
This review offers a comprehensive and in-depth overview of what is currently known regarding lactic acidosis in solid tumors. The authors have done an excellent job of succinctly and clearly explaining the most recent studies. Particularly notable is the examination of lactic acidosis' function in cancer cell metabolism and its potential as a therapeutic target. It offers a variety of fascinating new study possibilities. Overall, this abstract provides an important contribution to the understanding of cancer and establishes a clarifies for further research. I acknowledge your thoughtful and well-written article.
Comments
1-Patients with diabetes who are undergoing cancer treatment may also experience lactic acidosis. High blood sugar levels are a symptom of diabetes, which is a disease where the body is unable to metabolize and retain glucose effectively. These patients' cancer cells could be more reliant on glucose for energy and might create too much lactic acid, which would cause lactic acidosis. Please also discuss this in the text.
2- Alcoholism: The effects of alcohol on the liver cancer and other organs may cause those who misuse alcohol to develop lactic acidosis, which also has to be included in the text.
3-It has been shown that lactic acidosis contributes to the development of cancer treatment resistance in many ways. Here are a few ways that lactic acidosis might contribute to cancer medication resistance that should be included in the manuscript.
a- The prevention of apoptosis Apoptosis, a kind of programmed cell death, has been demonstrated to be inhibited by lactic acidosis in cancer cells. As cells that are resistant to apoptosis are less likely to be destroyed by chemotherapy or radiation treatment, this may contribute to cancer medication resistance.
b- p53 pathway, which is important in cell survival and drug resistance, is one of the signaling pathways that lactic acidosis has been reported to affect in cancer cells
Author Response
The referee has raised two relevant comments about the response of cancer cells to lactic acidosis and pathophysiological conditions like diabetes and alcoholism. These conditions further disrupt energy homeostasis and metabolism in cancer cells, adding further layers of metabolic complexity to the discussion. While we acknowledge that these conditions would be truly fascinating to discuss in the context of cancer cell metabolism, we prefer to focus the review on a general discussion in order to emphasize the critical metabolism-regulating role of lactic acidosis in cancer cell survival under glucose-depleted conditions. However, we do recognize the significance of external conditions in cancer cell behavior, and recognise this would create an excellent topic for a further review that would focus on the complexity of cancer metabolism.
In order to mention the targeting of lactic acidosis in other contexts than cancer, we replaced line 284: "Deciphering the critical role of lactic acidosis in favouring cancer cell survival in the tumour microenvironment has paved the way to several anticancer therapy strategies." by "Nowadays, lactic acidosis per se is targeted in therapies directed against cancer. Of note, it is also a major target in the treatment of type 2 diabetes (Aoi 2013, Gillies 2019)". To highlight that lactic acidosis promotes resistance to cancer therapies, we added line 36: "(Lactic acidosis is a hallmark of the tumour microenvironment) that has been shown to promote cancer resistance to chemotherapy (Barnes 2020, [47])".